# Survival of Recombinant Monoclonal Antibodies (IgG, IgA and sIgA) Versus Naturally-Occurring Antibodies (IgG and sIgA/IgA) in an Ex Vivo Infant Digestion Model

**DOI:** 10.3390/nu12030621

**Published:** 2020-02-27

**Authors:** Jiraporn Lueangsakulthai, Baidya Nath P. Sah, Brian P. Scottoline, David C. Dallas

**Affiliations:** 1Nutrition Program, School of Biological and Population Health Sciences, College of Public Health and Human Sciences, Oregon State University, Corvallis, OR 97331, USA; lueangsj@oregonstate.edu (J.L.); baidya.sah@oregonstate.edu (B.N.P.S.); 2Department of Pediatrics, Oregon Health and Sciences University, Portland, OR 97239, USA; scottoli@ohsu.edu

**Keywords:** palivizumab, immunoglobulins, infants, respiratory syncytial virus, gastrointestinal digestion

## Abstract

To prevent infectious diarrhea in infants, orally-supplemented enteric pathogen-specific recombinant antibodies would need to resist degradation in the gastrointestinal tract. Palivizumab, a recombinant antibody specific to respiratory syncytial virus (RSV), was used as a model to assess the digestion of neutralizing antibodies in infant digestion. The aim was to determine the remaining binding activity of RSV F protein-specific monoclonal and naturally-occurring immunoglobulins (Ig) in different isoforms (IgG, IgA, and sIgA) across an ex vivo model of infant digestion. RSV F protein-specific monoclonal immunoglobulins (IgG, IgA, and sIgA) and milk-derived naturally-occurring Ig (IgG and sIgA/IgA) were exposed to an ex vivo model of digestion using digestive samples from infants (gastric and intestinal samples). The survival of each antibody was tested via an RSV F protein-specific ELISA. Ex vivo gastric and intestinal digestion degraded palivizumab IgG, IgA, and sIgA (*p* < 0.05). However, the naturally-occurring RSV F protein-specific IgG and sIgA/IgA found in human milk were stable across gastric and intestinal ex vivo digestion. The structural differences between recombinant and naturally-occurring antibodies need to be closely examined to guide future design of recombinant antibodies with increased stability for use in the gastrointestinal tract.

## 1. Introduction

Infectious diarrhea is a leading cause of death in infants in developing countries [1,2]. Human milk contains naturally-occurring immunoglobulins IgG, IgA, and sIgA that can inhibit pathogen infections [3,4,5], and breastfeeding is protective against infectious diarrhea [6,7,8,9]. The World Health Organization reported that increasing breastfeeding rates by 40% worldwide for children less than 18 months of age would reduce diarrheal pathogen-induced deaths by 66% [10]. With human milk antibodies as a blueprint, oral supplementation of recombinant antibodies specific to enteric pathogens could enhance passive immunity in infants and prevent diarrhea. However, to work effectively, those antibodies would need to be stable and maintain their biological activity across the infant gastrointestinal tract. No previous studies have examined the stability of recombinant antibodies across the infant gastrointestinal tract. To gain insight into the feasibility of this concept, palivizumab, a recombinant humanized immunoglobulin G1 monoclonal antibody with high-binding affinity to the RSV (respiratory syncytial virus) fusion protein (F protein) and RSV neutralizing activity [11,12] was used as a model to study the stability of recombinant antibodies across the gastrointestinal tract. This antibody was selected as a model because it is the only pathogen-specific monoclonal antibody approved by the Food and Drug Administration (FDA) for use in high-risk infants (albeit for intramuscular injection). This FDA approval enabled us to gain ethical approval to feed this antibody to infants in a follow-on study to this one. We examined the digestion of palivizumab incubated in infant gastric and intestinal samples (ex vivo digestion). Ex vivo digestions are expected to better mimic in vivo digestion compared with in vitro digestion with simulated digestive fluids, because the digestive samples are the actual reaction conditions with a complete array of proteases and substances that would be present during in vivo digestion. To examine the potential stability of immunoglobulin isoforms in vivo, the digestion of a palivizumab idiotype in the setting of IgG, IgA, and sIgA isotypes was examined. The extent to which breastmilk-derived naturally-occurring RSV F protein-specific IgG and sIgA/IgA were degraded across ex vivo digestion was also examined. The aim of this study was to determine the remaining binding activity of palivizumab IgG, IgA, and sIgA and naturally-occurring IgG and sIgA/IgA across an ex vivo model of infant digestion. The data from this work can be used to inform the design of recombinant antibodies for gastrointestinal pathogen prevention in the future. 

## 2. Materials and Methods 

### 2.1. Sample Collection

This study was approved by the Institutional Review Boards of Oregon Health and Sciences University (STUDY 00018274). Infant inclusion criteria were the following: infants already admitted to the Doernbecher Children’s Hospital Neonatal Intensive Care Unit (NICU); greater than 34 weeks corrected gestational age (gestational age at birth plus days of life); an indwelling nasogastric or orogastric feeding tube; and tolerating full enteral feeding volumes. Infants were excluded from the study if they had anatomic or functional gastrointestinal disorders, were medically unstable, were nonviable, or had disorders that would be expected to affect normal digestion. After informed consent was obtained from parents, a nasal tube was placed into the distal duodenum or proximal jejunum, with the position of the sampling port confirmed by abdominal X-ray. After a normal feeding via the nasogastric tube, gastric and intestinal fluid samples were collected from three infant pairs (infant demographics are shown in Table 1): two infants born at 31 weeks and 6 days of gestational age (GA) were sampled at 33 and 34 days of life, and one born at 27 weeks and 1 day of GA was sampled at 76 days of life. The two infants were fed their mothers’ milk fortified with Similac Neosure®, and one infant was fed formula. These feeds were delivered via nasogastric tubes over 30 minutes or less. From the same feed, a volume (0.5–2 mL) of each infant’s gastric contents was collected 1 hour after the initiation of feeding in a 3 mL syringe back through the feeding tube via suction. Samples were collected from the nasojejunal/duodenal tube via gravity flow as the digesta passed the collection tube port. Gastric and intestinal samples were placed into sterile vials and stored at −80 °C. All samples were transported to Oregon State University on dry ice and stored at −80 °C upon arrival. Samples were thawed at 37 °C and separated into 100 μL aliquots.

### 2.2. Stability of Palivizumab IgG, IgA, and sIgA in Ex Vivo Gastric and Intestinal Digestion

Monoclonal antibodies with the palivizumab variable sequence were created as IgA and sIgA by the Center for Global Infectious Disease Research (CIDR, Seattle, WA, USA). The method for production is shown in the Appendix A. Palivizumab IgA was prepared by combining palivizumab IgA1 and IgA2 (18 mg/mL) in a 3:2 (v/v) ratio [13] (sequences are shown in Appendix A). Palivizumab sIgA was prepared by combining the secretory component (5.5 mg/mL) with palivizumab IgA (18 mg/mL) in a 2:1 (v/v) ratio and incubating for 30 minutes at room temperature (RT). Palivizumab (Synagis®) was purchased from MedImmune (Gaithersburg, MD, USA). One hundred microliter aliquots of gastric contents and intestinal contents from three infants were thawed rapidly at 37 °C. To evaluate the stability of the palivizumab IgG, IgA, and sIgA in gastric and intestinal contents, 25 μL of 400 μg/mL of each antibody in phosphate-buffered saline with 0.05% Tween-20 (PBST, Bio-Rad, Rockford, IL, USA) were combined with 75 μL of gastric or intestinal contents in low-binding Eppendorf tubes and mixed by pipetting. Immediately after mixing, ‘0 hour’ samples were collected as baselines. The gastric mixtures were incubated for 1 hour. Intestinal mixtures were incubated for 2 hours with 25 μL samples collected at 1 and 2 hours. Samples were incubated at 37 °C with shaking at 300 rpm using a thermomixer (Eppendorf, Hauppauge, NY, USA). All samples were stored at −80 °C immediately after collecting. 

### 2.3. Stability of Naturally-Occurring Milk RSV F Protein-Specific IgG and sIgA/IgA in Ex Vivo Gastric and Intestinal Digestion

To evaluate the stability of naturally-occurring RSV F protein-specific milk antibodies (IgG and sIgA/IgA) in gastric and intestinal contents, 100 μL aliquots of gastric and intestinal contents from the two infants that were fed mother’s milk were thawed rapidly at 37 °C. The samples were incubated at 0 and 1 hour (gastric contents) and at 0, 1, and 2 hours (intestinal contents) at 37 °C with shaking at 300 rpm using a thermomixer.

### 2.4. RSV F Protein-Specific ELISA

The palivizumab (IgG, IgA, and sIgA) and naturally-occurring RSV F protein-specific (IgG and sIgA/IgA) concentrations in gastric and intestinal samples and ex-vivo-incubated samples were quantified by ELISA as described by Mazur et al. [14] with modifications. Nunc MaxiSorp 96-well plates (Thermo Scientific, Waltham, MA, USA) were coated at 4 °C overnight with 100 μL of 100 ng/mL human RSV pre-fusion F protein (Sino Biological, Wayne, PA, USA). In between steps, plates were washed three times with 200 μL of phosphate-buffered saline with 0.05% Tween-20 using a Wellwash™ Versa microplate washer (Thermo Scientific). To prevent nonspecific binding of the antibodies, plates were blocked with 150 μL of 1% bovine serum albumin (Thermo Scientific) in PBST for 1 hour at RT. Samples were added (100 μL/well) in triplicate wells, at two preoptimized dilutions (400x and 800x for palivizumab IgG and naturally-occurring IgG, 800x and 1600x for palivizumab IgA, sIgA, and naturally-occurring sIgA/IgA), and incubated for 1 hour at RT. Palivizumab IgG, IgA, and sIgA were used to make a standard curve on each plate (at a range of 0–1000 ng/mL). One hundred microliters of 0.16 μg/mL goat anti-human IgG conjugated with horseradish peroxidase (Bio-Rad) were added for RSV F protein-specific IgG detection. One hundred microliters of 0.5 μg/mL goat anti-human IgA with horseradish peroxidase (Bio-Rad) were added for RSV F protein-specific IgA and sIgA detection. Plates were incubated for 1 hour at RT. The color was developed by adding 100 μL of 3,3′,5,5′-tetramethylbenzidine substrate (Thermo Scientific) for 5 minutes at RT. Fifty microliters of 2 N sulfuric acid were added to stop the reaction. Absorbance was measured at 450 nm with a spectrophotometer (SpectraMax M2, Molecular Devices). Data were interpreted using four-parameter logistic models to make the standard curve on each plate with R^2^ > 0.99 for goodness fit using Softmax® Pro 7.0 software. The concentration of each antibody measured by ELISA at 0 hour-incubation was denoted as 100% stability, and the percentage of stability at 1 hour and 2 hours post-incubation was calculated based on the measured concentration at each timepoint compared to the concentration at 0 hours. 

### 2.5. Validation of RSV F Protein-Specific Antibody ELISA

Gastric or intestinal samples (100 μL each) were supplemented with 100 μg/mL palivizumab (IgG, IgA, or sIgA). The RSV F protein-specific IgG, IgA, and sIgA ELISA were validated for precision, the lower limit of quantification (LLOQ), and upper limit of quantification (ULOQ) as described by Andreasson et al. [8] with some modifications. The RSV F protein-specific IgG, IgA, and sIgA ELISA were performed for gastric and intestinal samples on the same day with two dilutions and in triplicate. The precision of the assays was measured as the percentage coefficient of variation (CV), calculated from the following equation: % CV =Standard deviation ((SD))⁄(Average×100). The LLOQ was determined by identifying the lowest mean level of expected concentration above which the % CV < 20%, and the ULOQ by identifying the highest mean level of expected concentration below which the % CV < 20% for all samples. Based on this validation, only values that were above the LLOQ and below the ULOQ were used for sample quantification.

### 2.6. Statistical Analyses

One-way ANOVA followed by Tukey’s multiple comparison test (GraphPad Prism software, version 8.2.1) was applied to compare the percentage stability of antibodies in ex vivo intestinal contents. Unpaired *t*-tests were applied to compare the percentage stability of antibodies in ex vivo gastric contents. Differences were designated significant at *p* < 0.05. 

## 3. Results

### 3.1. Validation Parameters for RSV F Protein-Specific Antibodies ELISA

The RSV F protein-specific IgG, IgA, and sIgA ELISAs were validated (Table 2). Three gastric or intestinal samples spiked with either palivizumab IgG, IgA, or sIgA were analyzed within a single day with three replicates and two dilutions for each sample. In the gastric samples, the % CV of RSV F protein-specific IgG, IgA, and sIgA ELISAs were 19.59, 11.75, and 11.83, which meets typical validation requirements for assay precision (CV < 30%) [15]. In the intestinal samples, the % CV of RSV F protein-specific IgG, IgA, and sIgA ELISAs were 27.77, 13.86, and 15.25. In both the gastric and intestinal samples, the LLOQ of the RSV F protein-specific IgG, IgA, and sIgA ELISAs were 1, 25, and 5 ng/mL, respectively. The ULOQ of the RSV F protein-specific IgG, IgA, and sIgA ELISAs were 100, 250, and 250 ng/mL, respectively. The range of absorbance values for palivizumab IgG samples (diluted at 400x and 800x), palivizumab IgA and palivizumab sIgA samples (diluted at 800x and 1600x) were well within the linear range of each ELISA (between the LLOQ and ULOQ).

### 3.2. Survival of Palivizumab RSV F Protein-Specific IgG, IgA, and sIgA in Ex Vivo Gastric and Intestinal Digestion

The concentrations of palivizumab RSV F protein-specific IgG, IgA, and sIgA in ex vivo gastric and intestinal digestion were determined by the RSV F protein-specific ELISA (average concentrations are shown in Appendix A). Palivizumab IgG, IgA, and sIgA were degraded in both gastric and intestinal digestion (Figure 1A–F). Percentage stability of palivizumab IgG, IgA, and sIgA all decreased in 1-hour ex vivo gastric digestion (19.28%, 16.95%, and 20.50% decrease, respectively). Percentage stability of palivizumab IgG, IgA, and sIgA all decreased in 2-hour ex vivo intestinal digestion (35.71%, 20.59%, and 30.73% decrease, respectively). Percentage stability herein refers to the amount of antibody that is able to bind to the RSV F protein antigen and be bound by the detection antibody in the ELISA in the digestive sample in comparison with that in the feed. Our assumption is that this antigen and detection antibody binding capacity correlates with overall structural integrity and residual antibody functional (viral neutralization) capacity.

### 3.3. Survival of Naturally-Occurring Milk IgG and sIgA/IgA in ex vivo Gastric and Intestinal Digestion

The concentration of naturally-occurring RSV F protein-specific IgG and sIgA/IgA did not decrease after either gastric or intestinal digestion (Figure 2A–D). The extent of digestion did not differ between IgG and sIgA/IgA. The average concentrations of naturally-occurring RSV F protein-specific sIgA/IgA (3.24 ± 0.91 μg/mL) in the gastric and intestinal samples were 9.5-fold higher than that of IgG (0.34 ± 0.11 μg/mL) (Appendix A). The higher concentration of milk anti-RSV sIgA/IgA than of IgG matches previous findings that the total sIgA/IgA are in much higher concentrations than the total IgG in human milk [16,17]. 

## 4. Discussion

Infants have immature immune systems, and the ability of infants to fight infection is enhanced through the transmission of antibodies from the mother across the placenta and the presence of maternal antibodies in breastmilk to augment passive immunity [18,19]. Breastfeeding reduces the risk of enteric pathogenic diarrhea in infants [20], and mother’s milk contains antibodies specific to enteric pathogens that the mother has been exposed to previously [21]. Hypothetically, increasing the dose of enteric pathogen-specific antibodies to infants fed human milk could further decrease infection risk, and for those who are not fed human milk, provide enteral antibodies that they otherwise lack. Thus, the provision of supplemental oral recombinant antibodies is one potential strategy to enhance infant passive immunity. To be most effective in neutralizing pathogens present in the gastrointestinal tract, these recombinant antibodies would need to resist proteolysis and other structural degradation within the gut. However, the extent to which recombinant antibodies and naturally-occurring antibodies survive within an infant’s gastrointestinal tract remains unknown. Palivizumab, a humanized recombinant monoclonal anti-RSV F protein IgG1 that is FDA-approved for use in high-risk infants via an intramuscular route to provide passive immunity [22], was selected as a model to study the stability of recombinant antibodies across gastrointestinal digestion due to the ability to detect the protein in a complex mixture, the ability to assay for its activity, and its prior approval in infants, albeit via an alternate route. As a precursor to future in vivo digestion studies, in this study, palivizumab was incubated in infant gastric and intestinal samples. This ex vivo digestion was expected to mimic infant digestion as these samples contain all the enzymes and matrix compounds that would be present in an in vivo digestion. Using an ex vivo digestion approach enabled us to compare the digestion of isotypes of palivizumab (IgG vs. IgA vs. sIgA) without the complexities of gaining approval for in vivo administration of non-FDA-approved antibodies in infants, or the challenges of producing the larger amount of antibodies that would be expected to be needed for an in vivo feeding study. A subset of infants from which the gastric and intestinal samples were collected were fed human milk that contained naturally-occurring anti-RSV F protein IgG, IgA, and sIgA, and the degradation of these antibodies was also examined in those samples and compared with that of the recombinant antibodies. 

Palivizumab IgG, IgA, and sIgA were degraded across ex vivo gastric and intestinal digestion. After gastric and intestinal ex vivo digestion, the degree of digestion was similar between palivizumab isotypes. The optimum storage pH for palivizumab is 6.0 (Synagis® liquid solution); however, adjustment of the pH of milk spiked with palivizumab to pH 4, 7, and 8 followed by 2 hours incubation did not significantly decrease palivizumab stability (this method is shown in Appendix A and results are shown in Appendix A). These pH alterations match that of the infant gastric and intestinal samples. Therefore, it is most likely that the observed degradation of palivizumab in the stomach and intestine are the result of proteolytic degradation rather than pH-induced secondary or tertiary structural alteration. Naturally-occurring antibodies were not degraded across ex vivo digestion, and percentage stability across ex vivo digestion did not differ between isotypes for the naturally-occurring antibodies (IgG vs. sIgA/IgA). Naturally-occurring antibodies may be more resistant to ex vivo infant digestion than recombinant antibodies because of different structural features such as differing glycosylation profiles [23,24] and differing variable region amino acid sequences (polyclonal vs. monoclonal).

A limitation of this study is that the digestive samples were provided by only preterm infants, and thus, whether the digestion of these antibodies would differ in term infants remains unknown. As some data indicate that preterm infant protein digestion capacity is lower than that of term infants [25,26], particularly for milk immunoglobulins [16], a study with term infant digestive fluids would likely reveal increased digestion for each antibody isotype. 

The reduction of palivizumab IgG, IgA, and sIgA across ex vivo infant digestion highlights the need for continued work to improve recombinant antibody stability for potential use as oral supplements. Such antibodies could still be effective if the supplemental dose were increased to account for digestive losses. Adapting the structure of recombinant antibodies to be more similar to the digestion-resistant naturally-occurring antibodies would potentially also be advantageous. Methods for recombinant antibody production that include human-like glycosylation are now available [23,27,28,29]. Such methods may be important for enhancing the efficacy of recombinant antibodies in the prevention of enteric pathogen infections. 

## Figures and Tables

**Figure 1 nutrients-12-00621-f001:**
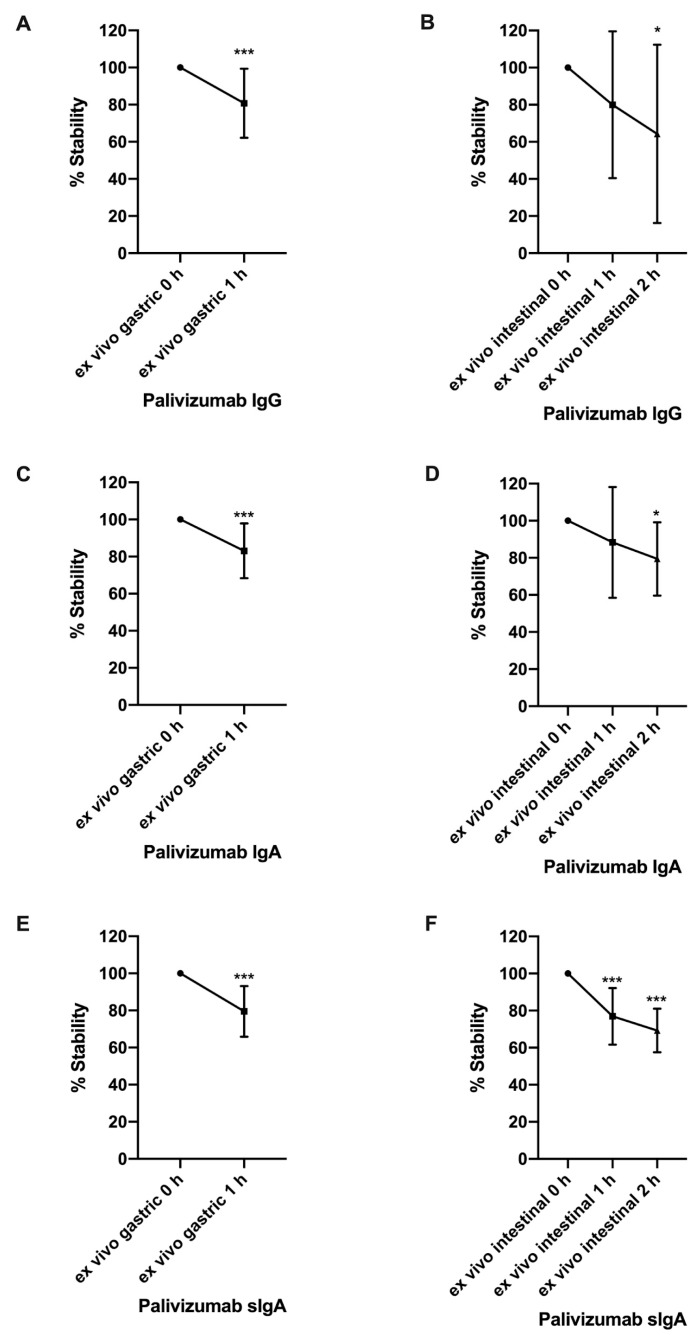
Percentage stability of palivizumab IgG, IgA, and sIgA across ex vivo infant digestion. (**A**) Percentage stability of palivizumab IgG, (**C**) IgA, and (**E**) sIgA across gastric digestion. (**B**) Percentage stability of palivizumab IgG, (**D**) IgA, and (**F**) sIgA across intestinal digestion. Values are mean ± SD, *n* = 18; three infants with two dilutions in triplicate. Asterisks represent the *p*-value (* *p* < 0.05 and *** *p* < 0.001) using the unpaired student’s *t*-tests to compare stability between gastric 0 h and 1 h samples. One-way ANOVA followed by Tukey’s multiple comparison tests were used to compare stability among intestinal 0, 1, and 2 h samples.

**Figure 2 nutrients-12-00621-f002:**
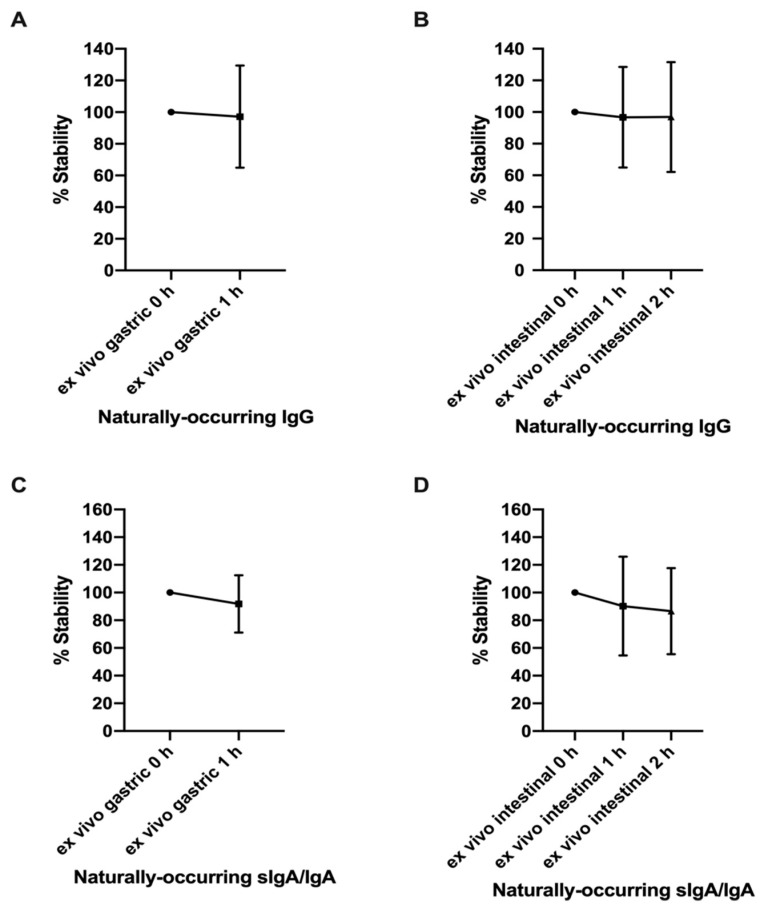
Percentage stability of naturally-occurring IgG and sIgA/IgA across ex vivo infant digestion. (**A**) Percentage stability of naturally-occurring IgG and (**C**) sIgA/IgA across gastric digestion. (**B**) Percentage stability of naturally-occurring IgG and (**D**) sIgA/IgA across intestinal digestion. Values are mean ± SD, *n* = 12; two infants with two dilutions in triplicate. The unpaired student’s *t*-tests were used to compare stability between gastric 0 h and 1 h samples. One-way ANOVA followed by Tukey’s multiple comparison tests were used to compare stability among intestinal 0, 1, and 2 h samples.

**Table 1 nutrients-12-00621-t001:** Demographics of preterm-delivering mother–infant pairs sampled for mother’s own breastmilk, formula milk, gastric contents (1 h postprandial time), and intestinal contents (2 h postprandial time).

Demographics	Infant 1	Infant 2	Infant 3
Gestational age (GA) at birth, weeks	31.6	31.6	27.1
Corrected Gestational age, weeks	36.2	36.3	38
Postnatal age at feeding, days	33	34	76
Bodyweight, kg	2.45	2.82	2.7
Length, cm	47	42	45
Head circumference, cm	35	35.5	32
Total kilocalories intake, kcal/kg/day	147	165	120
Feed sources	Mother’s milk fortified ^1^	Mother’s milk fortified ^1^	Formula

^1^ Mother’s milk fortified with Similac Neosure®.

**Table 2 nutrients-12-00621-t002:** Validation of RSV (respiratory syncytial virus) F protein-specific palivizumab IgG, IgA, and sIgA using the calculation of precision, lower limit of quantification (LLOQ), and upper limit of quantification (ULOQ).

Parameters	Palivizumab IgG		Palivizumab IgA		Palivizumab sIgA	
	Gastric	Intestinal	Gastric	Intestinal	Gastric	Intestinal
Precision (% CV)	19.59	27.77	11.75	13.86	11.83	15.25
LLOQ (ng/mL)	1	1	25	25	5	5
ULOQ (ng/mL)	100	100	250	250	250	250

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
