# Peer review of "Survival of Recombinant Monoclonal Antibodies (IgG, IgA and sIgA) Versus Naturally-Occurring Antibodies (IgG and sIgA/IgA) in an Ex Vivo Infant Digestion Model"

_nutrients, 2020, doi:10.3390/nu12030621_

Round 1
Reviewer 1 Report
Dear authors,
the study is interesting, well done, and the manuscript well written with good introduction and discussion. Only a few comments, the number of samples can be increased and could you reduce a little materials and methods section.
Reviewer 2 Report
In this article, the authors compared recombinant monoclonal (Palivizumab) and naturally occurring (milk-derived) immunoglobulins (Igs). The authors challenged the Igs by exposing them to an ex-vivo model of infant digestions that mimic both gastric and intestinal digestion, using digestive samples from infants. After the ex-vivo digestion, the antibodies were tested by a Respiratory syncytial virus (RSV) F protein-specific ELISA. Their results indicate that Palivizumab failed to survive the challenge, while milk-derived antibodies were stable, then successful to survive the challenge. The manuscript is well written, overall the research is interesting and of worldwide relevance as RSV is a virus of major impact. However, I have some general and specific points that I would recommend the authors to include or at least mention in order to enhance the quality of the manuscript and also highlight the importance of the research that they have done here. Also, I would recommend to re-submit this article for a new revision process as some points in the introduction, title and methodology are not entirely well connected.
General:
The manuscript uses the respiratory syncytial virus (RSV) in the title, which gives the impression that this research has something to do with illness caused by RSV. This virus is widely known to be the causative agent of mild upper respiratory tract illness, paediatric bronchiolitis and pneumonia. However, in the introduction section and along the entire manuscript, the authors focus their attention on diarrhoea, which is a symptom that may be associated, but not really the most important sign of RSV infection. In this sense, are the authors trying to make any link at all between RSV a diarrhoea? If not, I would recommend removing the virus name from the title, otherwise it gives the impression that the authors at some point will say something to do with RSV, but this research focus entirely on the immunoglobulins only. I would perhaps leave a simpler title such as: Differences in binding activity of commercially available monoclonal versus naturally occurring antibodies after ex vivo infant digestion model.In the introduction section, it should state clearly that these monoclonal antibodies were not manufactured at all to be administered orally and that the authors are performing this experiments in order to have some insights on features that monoclonal antibodies should have in the future such as resistance to low pH in order to protect against viruses that affect the digestive tract such as rotavirus, for instance. If, on the other hand, the authors are trying to make a link between RSV and diarrhoea, which I doubt, they should be able to state that clearly in the introduction.
There are interesting reasons why the development of vaccines against RSV has been hampered. (Please, review or at least mention in the article some iconic/catastrophic cases during the 60s). Please, explain why the development of vaccines might not be the best option and, perhaps that could explain better why a research like this one is of major importance, if it happens that RSV is the focus of this research, which again I doubt.
Along the manuscript, the authors use three concepts that seems to pin-point the same issue, however, I have concerns that they are actually synonyms. They first say that they aim to measure the ‘remaining binding activity of immunoglobulins’ (Line 18). Then, they say that they measure ‘the survival of each antibody’ (Line 22). Afterwards, the authors described in their materials and methods the ‘stability’ of different antibodies (Line 84 and 101). From my perspective, binding activity, antibody survival and stability are kind of the same aspects, but I am not sure that they are actually using the correct words for describing binding activity, which is what they are measuring with the ELISA test. Additionally, I doubt that all antibodies that survived to the digestion experiment are 100% stable, furthermore I doubt that all the stable ones have 100% binding activity and even more…How many of those that actually have binding activity (measured by ELISA) are actually protective against RSV infection? Could the authors enlarge their discussions on this?. Also, is it correct to have stability on the y-axis on the graphs? What about binding activity ?
I would recommend the authors to at least mention that palivizumab is usually administered by using the intramuscular route rather than oral. Also, would be interesting to know exactly at what pH the recombinant antibodies losses their stability. Do the authors have any thoughts on how muscle fatigue and, consequently a lower pH in the muscle could interfere with the monoclonal antibody stability? Could the authors maybe add an additional experiment testing different pH to determine specifically at what pH the antibodies losses their stability/binding activity/survival? Are there any other monoclonal antibodies they could test and maybe determine that monoclonal antibodies resist up to a certain pH only. Perhaps this information would be more relevant for future research regarding monoclonal antibodies against other viruses that indeed affect the digestive tract such as rotaviruses, for instance.
In the Material and Methods, they described well the methodology for sampling for the digestive content. However, I am missing how did the authors obtain the naturally occurring antibodies?
Reviewer 3 Report
1.- The title is too long and might be shortened (without abbreviates if posible)
2.- The Abstract is well done, but it is not structured. Please specify.
3.- The Introduction is nice, but has only 5 references that must be increased until ten at least.
4.- The babies were all prematures with clear differences in age, specially in the infant number 3, and postnatal age was longer also in infant 3 (Table 1 ). The secretion of the intestinal juice is not comparable between them. Please justify this selection and the comparability of all of three babies
5.- Section 2.1: Sample collection.
The 3 children were different. All of them are prematures and the composition of gastric and intestinal juices are not comparable with a baby born at term. Specially the third is under-weight and very immature. Please explain the reasons of this selection and the importance of the prematurity on the results of this study. The intake of child 3, was lower than the other two.
6.- Section 2.2. until 2.6 :No comments to add.
7.-Lines 225-227 :
Why the IgA/sIgA concentrations of RSV F proteins are 9.5 times higher in the intestinal fluid, than the corresponding IgG?.
Please give some explanations or comments of that
8.- Is different the efficacy of Palivizumab applied through intramuscular route than acting directly in the gastric and intestinal fluids?
9.- The discussion is well done and the conclusions of the study are clearly specified
10.- Tables and Figures are clear
11.- The references are very few (17). They must be increased until 25-30 in total